# Identification and Expression Profiling of Circulating MicroRNAs in Serum of *Cysticercus pisiformis*-Infected Rabbits

**DOI:** 10.3390/genes12101591

**Published:** 2021-10-09

**Authors:** Guoliang Chen, Liqun Wang, Tingli Liu, Yanping Li, Shaohua Zhang, Hong Li, Xuenong Luo

**Affiliations:** State Key Laboratory of Veterinary Etiological Biology/Key Laboratory of Veterinary Parasitology of Gansu Province, Lanzhou Veterinary Research Institute, Chinese Academy of Agricultural Sciences, Lanzhou 730046, China; glchen2019@163.com (G.C.); wlq1282690114@163.com (L.W.); LTL1114@163.com (T.L.); lyyp223@163.com (Y.L.); zhangshaohua@caas.cn (S.Z.); lihong16602307354@163.com (H.L.)

**Keywords:** serum miRNAs, *Cysticercus pisiformis*, novel-miR1, diagnostic targets

## Abstract

*Cysticercus pisiformis* (*C. pisiformis*), the larval form of *Taenia pisiformis*, parasitize mainly the liver, omentum and mesentery of rabbits and cause huge economic losses in the rabbit breeding industry. MicroRNA (miRNA), a short non-coding RNA, is widely and stably distributed in the plasma and serum. Numerous data demonstrates that, after parasitic infection, miRNAs become the key regulatory factor for controlling host biological processes. However, the roles of serum miRNAs in *C. pisiformis*-infected rabbits have not been elucidated. In this study, we compared miRNA expression profiles between the *C. pisiformis*-infected and healthy rabbit serum using RNA-seq. A total of 192 miRNAs were differentially expressed (fold change ≥ 2 and *p* < 0.05), including 79 up- and 113 downregulated miRNAs. These data were verified by qRT-PCR (real time quantitative polymerase chain reaction) analysis. Additionally, GO analysis showed that the target genes of these dysregulated miRNAs were most enriched in cellular, single-organism and metabolic processes. KEGG pathway analysis showed that these miRNAs target genes were involved in PI3K-Akt, viral carcinogenesis and B cell receptor signaling pathways. Interestingly, after aligning clean reads to the *T. pisiformis* genome, four (miR-124-3p_3, miR-124-3p_4, miR-124a and novel-miR1) *T. pisiformis*-derived miRNAs were found. Of these, novel-miR1was upregulated in different periods after *C. pisiformis* infection, which was verified qRT-PCR, and pre- novel-miR-1 was amplified from the cysticerci by RT-PCR, implying novel-miR-1 was derived from *C. pisiformis* and has great potential for the diagnosis of Cysticercosis pisiformis infection. This is the first investigation of miRNA expression profile and function in the serum of rabbits infected by *C. pisiformis*, providing fundamental data for developing diagnostic targets for Cysticercosis pisiformis.

## 1. Introduction

*Cysticercus pisiformis* (*C. pisiformis*), the larval form of *Taenia pisiformis* (*T. pisiformis*), is a heteroxenous parasite and belongs to Taenidae [1]. *T. pisiformis* parasitizes mainly the small intestines of canines, while *C. pisiformis* parasitizes mainly the omentum, liver and mesentery of lagomorphs and rodents [1,2,3]. *T. pisiformis* and *C. pisiformis* are distributed all over the world. In Cairo, Egypt, the prevalence of *T. pisiformis* infection in dogs was 70.8% [4]; in northern Italy, the cysticercosis of brown hares has been reported to be 3.28–14.8% [2]. However, in the state of Morelos, Mexico, the infection rate of hare cysticercosis is as high as 67.7% [5]. In China, *T. pisiformis* has also been reported [6]. Once infected with *C. pisiformis*, rabbits will show decreased prolificacy, weight loss and weakened immunological resistance, causing serious economic losses to the rabbit breeding industry [7,8]. The Dot-ELISA diagnostic method has been reported for detection of *C. pisiformis*, but has high false positive rate limit applicability [9]. Multiplex PCR has been studied as a diagnostic method for *T. pisiformis* in dog feces, but cannot be used for the diagnosis of rabbit cysticercosis [10]. Thus, there is an urgent need to develop a new method for early diagnosis for rabbit in order to reduce the losses due to cysticercosis.

MiRNAs are a class of endogenous non-coding RNA ranging in length from 18–24 nt [11]. Owing to the short length, miRNAs exist stably in various biosamples. miRNAs negatively regulating the expression of coding RNA at the post-transcriptional level have demonstrated that miRNAs play important roles in many diseases [12,13,14]. Some differentially expressed miRNAs have been proven as diagnostic targets in various diseases [15,16]. Numerous studies have shown that complex immune responses against parasites is activated in the host [17,18,19]. Of them, miRNAs have important functions in host–parasite interactions. For example, in *Cryptosporidium parvum* infection mouse model, the host-derived miR-27 suppressed the expression of KH-type splicing regulatory protein (KSRP) to promote iNOS mRNA stability and activate the TLR4/NF-κB signaling indirectly, which is involved in the epithelial antimicrobial defense process [17]. miR-101b-3p, expressed in mouse brain tissue infected with *Angiostrongylus cantonensis* (*A. cantonensis*), can suppress extracellular superoxide dismutase 3 (Acsod3) expression, thus increasing miR-101b-3p expression in mice that can ameliorate the oxidative damages of larvae [20]. miR-155-5p was shown to be involved in the central nervous system (CNS) inflammation induced by *A. cantonensis* infection [21]. Additionally, 58 host-origin serum miRNAs were significantly differentially expressed in mice infected by *E. multilocularis* [22]. Taken together, the above studies reveal the key role of miRNAs in the host response against parasitic infection and imply that dysregulated miRNAs could be involved in the biological processes of parasitic diseases.

Parasite-derived circulating miRNAs have shown great potential as diagnostic targets for some diseases. For instance, sja-miR-2b-5p and sja-miR-2c-5p were found to be significantly differentially expressed in patients infected by *Schistosoma japonicum* (*S. japonicum*) and can be used as biomarkers for the detection of human *S. japonicum* infection [23]. The differential expression of egr-miR-71 and egr-let-7 in the serum of cystic echinococcosis patients can also be promising biomarkers for early detection of hydatid cyst infection and post-surgical follow-up [24].

However, to the best of our knowledge, there is no data regarding the serum miRNA expression profile in rabbits infected with *C. pisiformis*. In the present study, we analyzed the miRNA expression profiles in *C. pisiformis*-infected and uninfected rabbit serum. Some dysregulated host and *C. pisiformis*-derived miRNAs were found and analyzed, which will improve understanding of the interactions between rabbits and *C. pisiformis* and provide fundamental miRNA data for screening serological diagnostic targets of cysticercosis pisiformis. 

## 2. Materials and Methods

### 2.1. Ethics Statement

This study was approved by the Animal Ethics Committee of Lanzhou Veterinary Research Institute, Chinese Academy of Agricultural Sciences (Permit No. LVRIAEC-2016-006), and the protocols used were strictly in accordance with the guidelines of animal welfare. The minimum amount of harm to experimental rabbits was required in the present study.

### 2.2. Parasites and Animals

*C. pisiformis* was isolated from naturally infected rabbits at a slaughterhouse in Jiaozuo city, Henan province, China. To obtain abundant *C. pisiformis*, each of the three 2-month-old pathogen-free dogs was orally challenged with 20 mature *C. pisiformis*. At 60 days post infection (PI), the eggs of *T. pisiformis* were collected from the feces of the *C.*
*pisiformis*-infected dogs, diluted with PBS to 1000 eggs/ml. Subsequently, six 1.5-month-old pathogen-free New Zealand white rabbits were randomly assigned into infection groups (Cpi groups, which including Cpi-1, Cpi-2 and Cpi-3) and control groups (NC groups, which included NC-1, NC-2 and NC-3). Each rabbit in the Cpi group was orally infected with 1 mL (1000 eggs) mature *T. pisiformis* eggs while the NC group was orally treated with 1 mL PBS. The rabbits in both groups were reared separately with forage and adequate lukewarm boiled water. The successful infection was determined by autopsy and examination of the number of *C. pisiformis*.

### 2.3. Blood Samples Collection and RNA Isolation

Fresh blood was collected from the rabbits in the Cpi and NC groups at 60 days PI and the total RNA was isolated from the separated serum samples using Trizol Reagent (Invitrogen, Carlsbad, CA, USA) according to the manufacturer’s protocol. Furthermore, the six rabbits were necropsied to confirm infection by *C. pisiformis*. The results of the examination are shown in Table 1.

### 2.4. Small RNA Library Generation and Sequencing

Small RNA fragments were separated from the six total RNA samples by polyacrylamide gel electrophoresis (PAGE), and RNA with the lengths of 18–30 nt was collected. Subsequently, 5-adenylated and 3-blocked adaptors were ligated to the 3′ end of the small RNA fragments, then unique molecular identifiers (UMI) [25] labeled primers were added to hybridize with the 3′ adaptor. Unlinked adaptors were enzyme digested after 5′ end adaptor ligation. cDNA first strand was synthesized using RT primers for the UMI. The cDNA library containing both 5′ and 3′ adaptors were amplified by PCR and products 100–120 bp in length were selected to construct small RNA libraries. After library quantitation, pooling cyclization and quality control, the qualified libraries were prepared for RNA sequencing. The qualified sequences were sequenced on a BGISEQ-500 (Beijing Genomics Institute, China) and the sequences (raw reads) were produced for subsequent analysis.

### 2.5. Reads Analysis and MiRNA Prediction 

The clean reads were obtained after removing the low-quality reads (those with 5′ primer contaminants, without 3′ primer and/or without insertion) from the raw reads. The clean reads were mapped against the *Oryctolagus cuniculus* genome (https://www.uniprot.org/uniprot/?query=taxonomy:9986 (accessed on 19 December 2019)) and *T. pisiformis* genome (unreleased) using Bowtie2 software. The mapped reads were classified and annotated into different kinds of small RNAs (miRNAs, snRNA, snoRNA, Rfam other sncRNA, rRNA, tRNA). Of them, the miRNA maturity and precursor were selected by comparing against the known mature miRNA in the miRbase database Unknown tags were used to predict novel-miRNAs through miRDeep2 by exploring the characteristic hairpin structure of the miRNA precursor and the minimum free energy (MFE) of the candidate miRNA secondary structure [26,27].

### 2.6. Verification of Differential Expression MiRNAs by Using qRT-PCR

The miRNAs from the fresh blood samples were separated using an EasyPure miRNA Kit (TransGen, Beijing, China) according to the manufacturer’s instructions. Subsequently, the miRNAs were reverse transcribed into cDNA using the Mir-XTM miRNA First-Strand Synthesis Kit (Clontech, Shanghai, China). qRT-PCR was performed in a 25 μL reaction volume, including 12.5 μL SYBR Advantage Premix (2×) (Takara, Shiga-ken, Japan), 2 µL cDNA, 0.5 µL ROX Dye (50×), 0.5 µL miRNA-specific forward primer, 0.5 µL mRQ 3′ primer and 9 µL RNase free water. qRT-PCR was performed with the following conditions: 95 °C for 10 min and 40 cycles of 95 °C for 15 s, 60 °C for 1 min. The expression of 8 target miRNAs in rabbit serum at two months PI with *T. pisiformis* were normalized to U6 small nuclear RNA (snRNA), with the primers (Sangon Biotech, Shanghai, China) listed in Table 2. The 2^−ΔΔCt^ method was used to calculate the relative expression of candidate miRNAs [28].

### 2.7. Function Prediction of Dysregulated MiRNAs

To explore the potential functions of differentially expressed miRNAs (DEMs), target genes were predicted using RNAhybrid [29] and miRanda [30] software. The target genes were functionally annotated by assigning Gene Ontology terms (GO-terms, http://www.geneontology.org/ (accessed on 29 November 2019)) [31] and KEGG pathway analysis [32]. By calculating the number of genes per-term and applying the hypergeometric test, the DEM target gene functions were subjected to GO function classification. Similarly, KEGG was used to perform pathway enrichment analysis.

### 2.8. Statistical Analysis

Student’s *t*-test was used to analyze the significance of miRNA expression and Graphpad 7 was used to create the relative expression diagram. *p* < 0.05 was considered statistically significant.

## 3. Results

### 3.1. Characterization of the sRNAs Libraries

Among the six small RNA (sRNAs) libraries, the average raw tags were 28,000,000 per library. After removing low-quality tags in each library, about 26,000,000 clean tags were obtained, which accounted for greater than 90% of the raw tags. Clean tags were aligned to the reference genome of *Oryctolagus cuniculus* (https://www.uniprot.org/uniprot/?query=taxonomy:9986 (accessed on 19 December 2019)) and approximately 23,000,000 tags were mapped for mapping rates between 85.11% to 94.11%. In order to determine whether or not there were sRNAs from *C. pisiformis*, we mapped the clean tags to the *T. pisiformis* genome (unreleased). The result showed that 1,337,063 tags were aligned to the *T. pisiformis* genome for mapping rates from 2.6% to 5.27%. The specific data of each group are shown in Table 1.

The tags which mapped to the *Oryctolagus cuniculus* genome were annotated into different kinds of sRNAs (miRNAs, snRNA, snoRNA, Rfam other sncRNA, rRNA, tRNA), genes (exon, intron, intergenic), unmapped and repeat sequences. Of them, 3.15% were classified as miRNAs (mature, hairpin and precursor) (Figure 1a). When mapped to the *T. pisiformis* genome, the proportion of miRNAs was 6.17% (Figure 1b). Additionally, sRNA populations of between 17 and 26 nt in length (with the most abundant length being 22 nt and 23 nt) were observed in both species (Figure 2), which is consistent with miRNA common length.

It is worth mentioning that in the present study, the results of Cpi-3 and NC-3 samples were quite different from the other samples. To obtain accurate results, we analyzed only two sets of data of Cpi-1/ Cpi-2 and NC-1/ NC-2 in the next study.

### 3.2. Identification of Differentially Expressed miRNAs

By comparison, a total of 530 rabbit-derived miRNAs were found to be differentially expressed in the Cpi-1 and NC-1 groups (Appendix A), and 548 in the Cpi-2 and NC-2 groups (Appendix A). Among them, 192 duplicated miRNAs (71 upregulated and 121 downregulated;|log2(Cpi/NC)| > 1 and FDR < 0.001; Appendix A) were significantly differentially expressed in the two infected groups. Of these 192 miRNAs, 20 known miRNAs and 172 novel predicted miRNAs were included. The sequences and reads of the 20 known miRNAs are shown in Figure 3.

Interestingly, four deregulated *T. pisiformis*-derived miRNAs (miR-124-3p_3, miR-124-3p_4, miR-124a and novel-miR1) were found in the two Cpi groups (Appendix A). However, novel-miR1 was the only upregulated miRNA, which was verified by qRT-PCR at different time points after infection. The stem-loop structure for the putative pre-miRNAs of novel-miR1 was predicted by miRdeep2 and the sequence of pre-novel-miR1 was specifically amplified from the *C. pisiformis* (shown in Figure 4). Further analysis found that miR-124-3p_3, miR-124-3p_4, miR-124a were also present in the rabbit samples. The miRNA distribution of each group is shown in Figure 5.

### 3.3. qRT-PCR Verification of miRNA Expression

Eight differentially expressed miRNAs from the rabbits were randomly selected for qRT-PCR to confirm sequencing data. The results showed that the expression trend of all candidate miRNAs, seven upregulated (novel-mir128, novel-mir303, novel-mir307, novel-mir318, novel-mir476, novel-mir857 and novel-mir150) and one downregulated miRNA (novel-mir57) were consistent with the sequencing results, which support the accuracy of miRNA sequencing (Figure 6).

### 3.4. Prediction of miRNA Target Genes

In order to further identify the importance of miRNA/mRNA-regulated axes in *T. pisiformis* infection, RNAhybrid [29] and miRanda [30] were used to predict the DEM target genes. A total 2,202,871 genes were predicted as target genes of 886 differentially expressed rabbit miRNAs (Appendix A). A total of 140 target genes were predicted from four *T. pisiformis*-derived miRNAs (Appendix A). In order to confirm these findings more definitively, 10 rabbit DEMs (5 up- and 5 downregulated miRNAs) were selected to construct an interaction network using Cytoscape_3.7.2, which included 348 nodes and 412 edges (Figure 7a). Additionally, four *T. pisiformis*-derived miRNAs were used to construct an interaction network that included 140 nodes (Figure 7b). From the network diagram, it was clear that the target XM.008265418.2 (tyrosine kinase with immunoglobulin-like and EGF-like domains 1, TIE1) was regulated by the novel-miR388, novel-miR62 and novel-miR635. Additionally, the target genes of miR-124a such as Mark00009255, Mark00000443, Mark00007007 and Mark00009550 can be clearly seen through the network diagram.

### 3.5. Functional Analysis

To determine the roles of the rabbit DEMs, the top 10 terms (biological processes, cellular component and molecular function) in the Cpi and NC groups were obtained by gene ontology term analysis (Figure 8a). Among them, the most abundant GO terms under the biological process (BP) category were cellular process, single-organism process and metabolic process. The most enriched cellular component (CC) terms were organelle, cell part and membrane. Finally, molecular transducer activity, catalytic activity and binding were the most enriched molecular function (MF) terms. The KEGG pathway analysis of potential target genes were involved in some of the signaling pathways responsible for tight junction, PI3K-Akt signaling pathway, microRNAs in cancer, cell adhesion molecules and apoptosis (Figure 8b).

The target genes of *T. pisiformis* miRNAs enriched by GO terms of BP, MF and CC are shown in Figure 9a. The notable GO terms were cellular process, metabolic process and catalytic activity. KEGG pathway analysis showed that the majority of the target genes of the *T. pisiformis*-derived miRNA participated in viral carcinogenesis, osteoclast differentiation, herpes simplex infection and B cell receptor signaling pathway (Figure 9b).

## 4. Discussion

*T. pisiformis* is a heteroxenous parasite, which belongs to Platyhelminthes, Cestoda, Eucestoda, Cyclophyllidea, Taeniidae and Taenia taxonomy [2]. Rabbit infection with *T. pisiformis* result in significant economic losses and animals’ welfare deduction in the rabbit breeding industry [6]. *T. pisiformis* infection can decrease rabbit reproductive rates and induce mass death [5], and may lead to hare breed extinction [2,3]. There is no effective strategy to diagnose cysticercosis pisiformis, which can only be confirmed through autopsy. MiRNAs, as small non-coding RNAs, can regulate gene expression at the post-transcriptional level [33,34] and is closely related to pathogen infection. In the present study, we extracted the serum miRNAs of rabbits in Cpi and NC groups and identified 192 DEMs, including 79 upregulated miRNAs (1 known miRNA) and 113 downregulated miRNAs (19 known miRNAs). Of the 20 known miRNAs, 75% belong to 5 miRNA families: let-7 (let-7-5p, let-7-5p_3, let-7-5p_4), miR-29b (miR-29b, miR-29b-3p, miR-29b-3p_2), miR-10 (miR-10, miR-10-5p_2), miR-33 (miR-33-3, miR-33-5p_3), miR-7 (miR-7, miR-7-5p_4, miR-7-5p_6), miR-92 (miR-92, miR-92-3P_4). 25% of the known miRNAs were miR-100-2, miR-124-3p_4, miR-125-5p_4, miR-133-3p_2, miR-9-5p_2. Additionally, four *T. pisiformis* miRNAs (miR-124-3p_3, miR-124-3p_4, miR-124a and novel-miR-1) were found in the *T. pisiformis* genome. For further comparison, we found that miR-124-3p_4 existed not only in rabbit serum and *T. pisiformis*, but also in *C. pisiformis* exosome-like vesicles [35], showing that *C. pisiformis*-derived miR-124-3p_4 was carried into the host serum by exosome-like vesicles as cargo. A total of 414 target genes of miR-124-3p_4 was found by RNAhybrid and miRanda. The function or components of these target genes mainly include hypoxia associated factor, tryptophan hydroxylase, zinc finger SWIM domain containing protein, pyridoxal dependent decarboxylase, oxidoreductase, GPN loop GTPase 2 and kyphoscoliosis peptidase.

Other DEMs in *C. pisiformis*-infected rabbit serum, such as let-7-5p, miR-7, miR-9-5p and miR-125-5p_4, were also significantly differentially expressed in the *Hymenolepis microstoma* larvae [36], showing their important roles in the interactions between host and metacestode of tapeworms. The most abundant miR-10-5p and let-7-5p in *Taenia solium* and *Taenia crassiceps* characterized immunosuppressive effects on murine peritoneal macrophages, which strongly decreased the expression of interleukin 6 (IL-6), tumor necrosis factor (TNF) and interleukin (IL-12) secretion. They also moderately decreased nitric oxide synthase inducible (NOS2) and Il1b expression (pro-inflammatory cytokines) in M (IFN-γ) macrophages, expression of Tgf1b, and the secretion of IL-10 (anti-inflammatory cytokines) in M (IL-4) macrophages [37]. miRNA let-7-5p from *C. pisiformis* exosomes has been reported to decrease M1 phenotype expression while promoting polarization to the M2 phenotype through inhibition of the expression of transcription factor CCAAT/enhancer-binding protein (C/EBP)-δ. However, let-7-5p was also detected in rabbit serum at 2 months post infection, and may be released into rabbit serum through *C. pisiformis* exosomes [38].

In addition, the miRNAs were relatively stable in body fluids and tissues [39]. Many miRNAs have been proven as biomarkers with great potential in the diagnosis and treatment of infectious diseases and cancer [40,41,42]. Of course, there are many studies about miRNAs as biomarkers to diagnose or detect parasitic infection. For instance, miR-277, miR-3479-3p and bantam, in human serum from Schistosome endemic areas, were reported to detect infected *S*. *mansoni* individuals from low and high infection intensity sites with specificity/sensitivity values of 89%/80% and 80%/90%, respectively [43]. Additionally, parasite-derived miRNAs sja-miR-2b-5p, sja-miR-2c-5p, sja-miR-277 and sja-miR-3479-3p in mouse serum were shown to have potential as diagnostic markers for Schistosomiasis japonica [23,44]. Previous research has shown that hydatidosis can be diagnosed early and monitored through the biomarker of serum-derived egr-miR-71 [24]. The expression of hsa-miR-125b-5p was stably upregulated in the plasma and liver tissue samples from patients with alveolar echinococcosis (AE), which suggested that hsa-miR-125b-5p may be a promising biomarker for early non-invasive diagnosis of AE [45]. Serum aca-miR-146a was found to be expressed significantly higher in mice infected by *A**ngiostrongylus cantonensis* and its receiver operating characteristic (ROC) curve analysis showed that aca-miR-146a was an effective biomarker for discriminating the infected from uninfected cases with an area under the ROC curve (AUC) of 0.90. Its diagnostic accuracy was assessed on patients (*n* = 30) and healthy controls (*n* = 30), and the sensitivity and specificity reached 83% and 86.7%, respectively, indicating that the aca-miR-146a in serum is an effective biomarker to track infection of *A.*
*cantonensis* [46].

Furthermore, in the present study, we amplified the precursor sequence of novel-miR1 from *C. pisiformis* and found that novel-miR1 is unique to *C. pisiformis*. qRT-PCR analysis revealed that novel-miR-1 was upregulated in the serum of rabbits infected with *C. pisiformis* 2, 3 and 7 months after infection (Figure 4). As the candidate diagnosis target of Cysticercosis pisiform, novel-miR1 has a strong specificity and should be further investigated.

## 5. Conclusions

To the best of our knowledge, this study was the first to determine miRNA expression pattern in rabbits infected with *C. pisiformis*. A total of 192 DEMs were identified, including 20 known and 172 novel-miRNAs. Additionally, four *T. pisiformis* miRNAs were found to be differentially expressed, but only novel-miR1 was derived strictly from *C. pisiformis* and upregulated in different periods in rabbits infected with *T. pisiformis*. Further miRNAs data analysis showed that these deregulated miRNAs were most enriched in cellular processes, single-organism processes, metabolic processes and molecular transducer activity. In addition, the signaling pathways of those miRNA were involved in PI3K-Akt, viral carcinogenesis and B cell receptor signaling pathways. Most importantly, novel-miR1 shows promising diagnostic potential for cysticercosis pisiformis.

## Figures and Tables

**Figure 1 genes-12-01591-f001:**
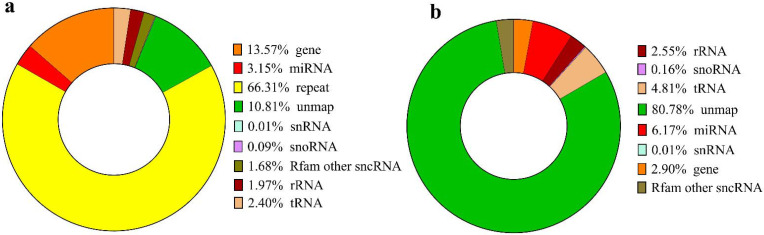
Classification and proportion of small RNA mapped tags. (**a**): classification and proportion of small RNA about the tags mapped in the *Oryctolagus cuniculus* genome; (**b**): classification and proportion of small RNA tags mapped in the *T. pisiformis* genome.

**Figure 2 genes-12-01591-f002:**
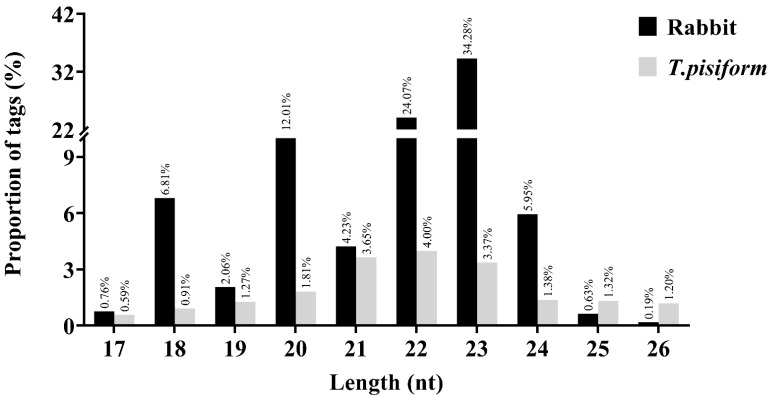
The proportion of the small RNA sequences in *C. pisiformis*-infected rabbit serum.

**Figure 3 genes-12-01591-f003:**
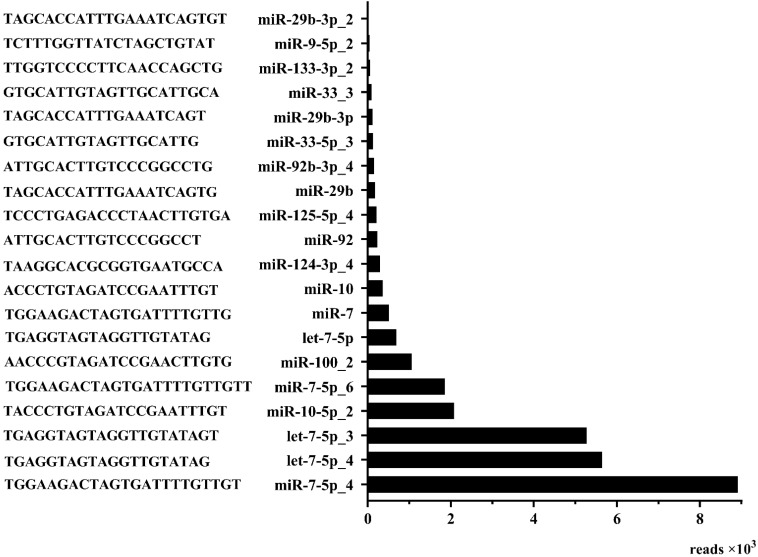
The sequences and reads of 20 significantly differentially expressed known miRNAs.

**Figure 4 genes-12-01591-f004:**
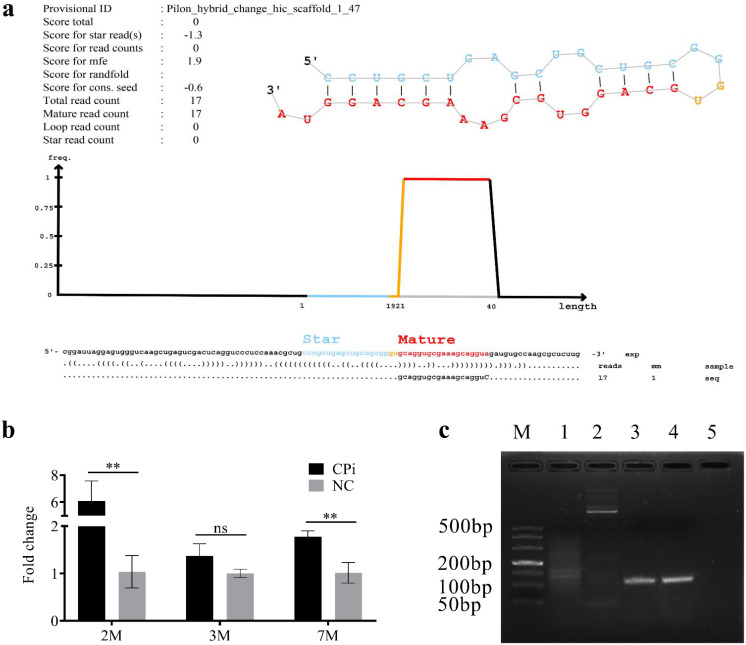
Information and test results of novel-miR1. (**a**): graph and statistics of novel-miR1 prediction by miRDeep2. Both star and mature strands were detected and integrated. Upper left table shows information about the sample and the miRDeep2 scores; upper right shows predicted stem-loop structures of the novel-miR1. The mature sequence is shown in red, and the star sequence is shown in blue; (**b**): qRT-PCR analysis of novel-miR1 expression by in the serum of rabbit infected with *Cysticercus pisiformis* 2, 3 and 7 months post-infection. The 2- and 7-month expression levels of novel-miR1 are significantly upregulated. The 3-month expression of novel-miR1 trended to be upregulated, **, *p* < 0.01 (Student’s *t*-test); c: PCR amplification results of the fragment containing novel-miR-1 precursor. M is the DL 500 DNA Marker, 1 and 2 are the PCR amplification results from rabbit serum, 3 and 4 are the PCR amplification results from *Cysticercus pisiformis* tissue and 5 is the negative control.

**Figure 5 genes-12-01591-f005:**
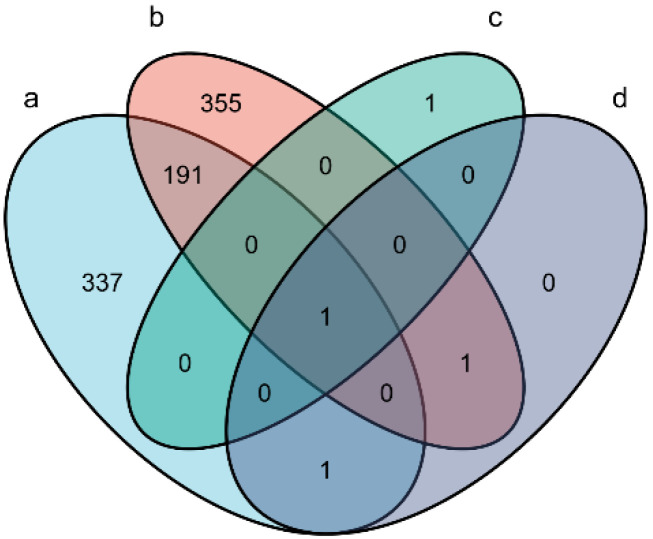
The distribution of differentially expressed miRNAs in each group. a: the differentially expressed rabbit miRNAs in the Cpi1-NC1 group. b: the differentially expressed rabbit miRNAs in the Cpi2-NC2 group. c: the differentially expressed *T. pisiformis* miRNAs in the Cpi1-NC1 group. d: the differentially expressed *T. pisiformis* miRNAs in the Cpi2-NC2 group. The overlapping region indicates duplicated differentially expressed miRNAs.

**Figure 6 genes-12-01591-f006:**
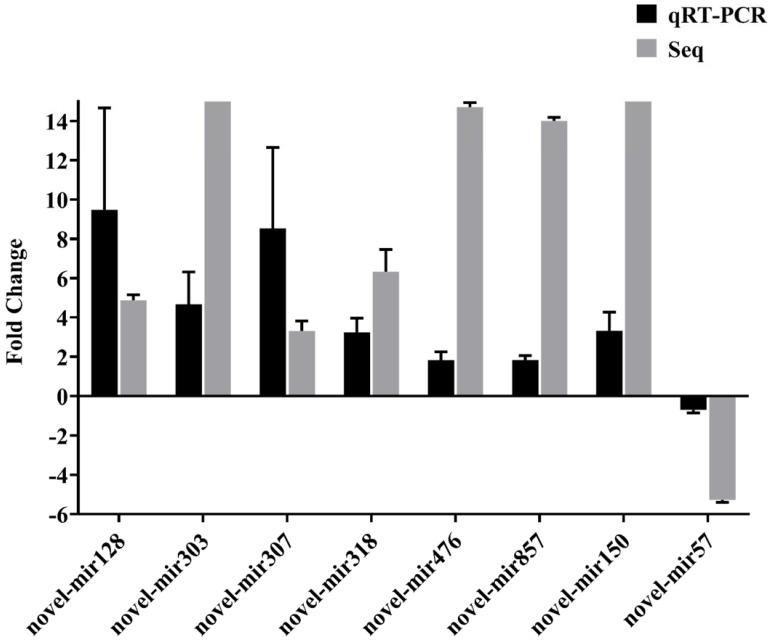
qRT-PCR validation of miRNA expression in the serum of *Cysticercus pisiformis* infected rabbits 2 months post-infection. Data for the final analysis were from three independent experiments.

**Figure 7 genes-12-01591-f007:**
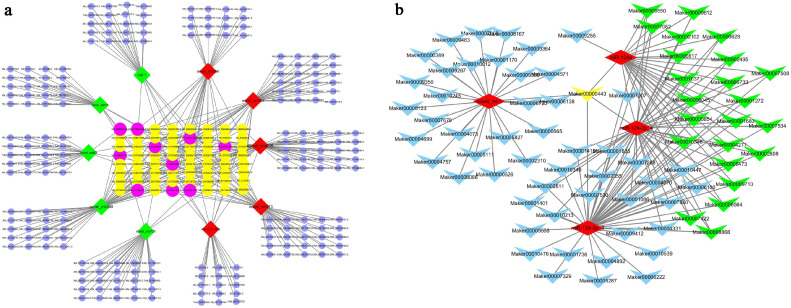
Network of the representative differentially expressed miRNA partial target mRNAs. (**a**): the differentially expressed rabbit miRNAs and their partial target mRNAs. Different colors are used to show different genes, with red for upregulated miRNAs and green for downregulated miRNAs. Blue, yellow and purple represent the mRNAs that were regulated by one, two and three miRNAs, respectively. A line indicates the connection of miRNAs and mRNAs. (**b**): the differentially expressed *T. pisiformis* miRNAs and their partial target mRNAs. Red represents miRNAs. Blue, green and yellow represent mRNAs that were regulated by one, two and three miRNAs, respectively. A line indicates the connection of miRNAs and mRNAs.

**Figure 8 genes-12-01591-f008:**
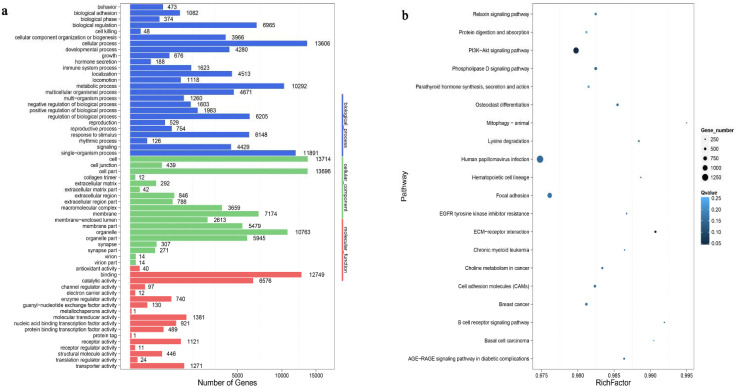
Function analysis of the target genes of differentially expressed rabbit miRNAs (**a**): GO molecular function annotations of the target genes of differentially expressed rabbit miRNAs. According to *p* value, the top GO terms biological processes, cellular component and molecular function are shown. (**b**): KEGG pathway analysis of predicted target genes of differentially expressed rabbit miRNAs. The number of genes is indicated by the size of dots.

**Figure 9 genes-12-01591-f009:**
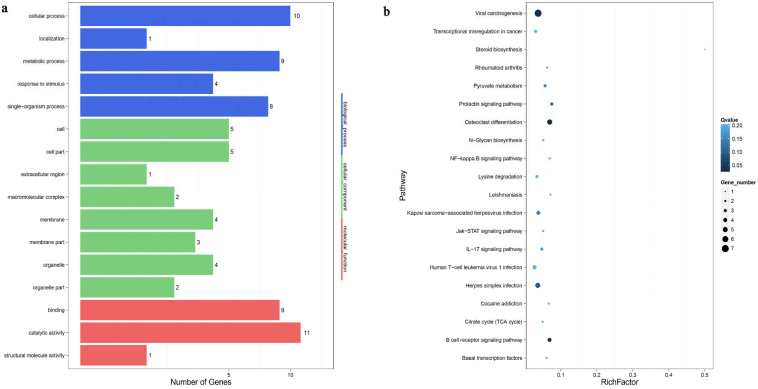
Functional analysis of the target genes of differentially expressed *T*. *pisiformis* miRNAs (**a**): GO molecular function annotations of the target genes of differentially expressed *T. pisiformis* miRNAs. According to *p* value, the top GO terms biological processes, cellular component and molecular function are shown. (**b**): the KEGG pathway analysis of predicted target genes of differentially expressed *T. pisiformis* miRNAs. The number of genes is indicated by the size of dots.

**Table 1 genes-12-01591-t001:** Characteristics of infection status and RNA-seq results.

Sample Name	Number of Infected *C. pisiformis*	Raw Tag Count	Clean Tag Count	Percentage of Clean Tag(%)	Mapped Tag of *Oryctolagus* *cuniculus* Genome	Percentage of Mapped Tag of *Oryctolagus* *cuniculus* Genome(%)	Mapped Tag of *T.* *pisiformis* Genome	Percentage of Mapped Tag of *T.* *pisiformis* Genome(%)	Known miRNA Count	Novel miRNA Count
NC-1	0	28,101,107	25,647,172	91.27	22,472,334	87.62	1,329,791	5.18	23	491
NC-2	0	28,011,580	25,355,821	90.52	23,862,803	94.11	1,337,063	5.27	40	186
NC-3	0	28,343,925	26,650,162	94.02	22,681,061	85.11	1,267,343	4.76	17	453
Cpi-1	180	27,813,172	25,843,308	92.92	22,232,314	86.03	671,980	2.60	39	216
Cpi-2	206	28,839,799	27,578,609	95.63	25,076,910	90.93	1,250,296	4.53	28	121
Cpi-3	194	28,139,567	26,404,625	93.83	23,154,753	87.69	788,071	2.98	27	112

**Table 2 genes-12-01591-t002:** Primer sequences for qRT-PCR.

miRNAs	Primers (5′-3′)
U6	F: GCTTCGGCAGCACATATACTAAAAT
	R: CGCTTCACGAATTTGCGTGTCAT
novel-miR1	TATACGCAGGTGCGAAAGCAGG
novel-miR857	AGAGAGCGGTCGGACACC
novel-miR476	TGATTGGTGAGCGTAGAGGTCG
novel-miR318	TATATAGTGGCGCGAAGCGGG
novel-miR307	TATATACCCGCGAGGGGGC
novel-miR303	TATATACCGGGGCGGGGTG
novel-miR128	TATATATAGCGCGCGTGCGCC
novel-miR150	TATATAGGCGGGGTGCGGG
novel-miR57	CGCAATTGCACGGTATCCATCTGT

## Data Availability

Data are contained within the article. For further enquiries contact the corresponding author at glchen2019@163.com.

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
