# Peer review of "Identification and Expression Profiling of Circulating MicroRNAs in Serum of Cysticercus pisiformis-Infected Rabbits"

_genes, 2021, doi:10.3390/genes12101591_

Round 1
Reviewer 1 Report
The manuscript entitled "Identification and Expression Profiling of Circulating MicroRNAs in Serum of Cysticercus Pisiformis Infected Rabbits " investigates miRNA expression expression profile and function in the serum of rabbits infected with C. pisiformis, providing fundamental data critical to the development of diagnostic targets for cysticercosis pisiformis. The paper is generally well written and structured. Sufficient information about the previous studies findings is presented for readers to follow the present study rationale and procedures.
I recommend some editing:
- all parasite names should be written in italics.
-the distribution of clinical groups is unclear. You should rewrite the section " 2.2. Parasites and Animals"
- Table 1 is unclear, it should be made more readable.
Author Response
Responses to comments and suggestions of Reviewer #1:
General comments: The manuscript entitled "Identification and Expression Profiling of Circulating MicroRNAs in Serum of Cysticercus Pisiformis Infected Rabbits " investigates miRNA expression expression profile and function in the serum of rabbits infected with C. pisiformis, providing fundamental data critical to the development of diagnostic targets for cysticercosis pisiformis. The paper is generally well written and structured. Sufficient information about the previous studies findings is presented for readers to follow the present study rationale and procedures.
Response: Thanks for your favorable and valuable suggestions on our MS, and we have improved the English quality and revised our MS strictly according to your suggestions.
Point 1: All parasite names should be written in italics.
Response: Revised accordingly.
Point 2: the distribution of clinical groups is unclear. You should rewrite the section " 2.2. Parasites and Animals"
Response: We have rewrited the section 2.2.
Point 3: Table 1 is unclear, it should be made more readable.
Response: Revised accordingly.

Reviewer 2 Report
The Authors have studied the miRNA expression profile and function in the serum of rabbits infected by C. pisiformis, providing fundamental data for developing diagnostic targets for Cysticercosis pisiformis the larval form of Taenia pisiformis, parasitize mainly the liver, omentum and mesentery of rabbits and cause huge economic losses in the rabbit breeding industry. The manuscript is very interesting and images quality is very good.
Author Response
Thank you for your favorable comments and valuable suggestions on our MS. We have improved the quality of English and revised our MS strictly in accordance with your suggestions.
